# The Effects of Time-Restricted Eating on Fat Loss in Adults with Overweight and Obese Depend upon the Eating Window and Intervention Strategies: A Systematic Review and Meta-Analysis

**DOI:** 10.3390/nu16193390

**Published:** 2024-10-05

**Authors:** Yixun Xie, Kaixiang Zhou, Zhangyuting Shang, Dapeng Bao, Junhong Zhou

**Affiliations:** 1College of Education, Beijing Sport University, Beijing 100084, China; 2022210556@bsu.edu.cn; 2College of Physical Education and Health Science, Chongqing Normal University, Chongqing 401331, China; 20230102@cqnu.edu.cn; 3College of Physical Education and Health Management, Chongqing University of Education, Chongqing 400065, China; shangzhang0301@163.com; 4China Institute of Sport and Health Science, Beijing Sport University, Beijing 100084, China; 5Hebrew SeniorLife Hinda and Arthur Marcus Institute for Aging Research, Harvard Medical School, Boston, MA 02115, USA; junhongzhou@hsl.harvard.edu

**Keywords:** time-restricted eating, fat loss, overweight, obesity, meta-analysis

## Abstract

Time-restricted eating (TRE) is a circadian rhythm-based intermittent fasting intervention that has been used to treat obesity. However, the efficacy and safety of TRE for fat loss have not been comprehensively examined and the influences of TRE characteristics on such effects are unknown. This systematic review and meta-analysis comprehensively characterized the efficacy and safety of TRE for fat loss in adults with overweight and obese, and it explored the influence of TRE characteristics on this effect. Methods: A search strategy based on the PICOS principle was used to find relevant publications in seven databases. The outcomes were body composition, anthropometric indicators, and blood lipid metrics. Twenty publications (20 studies) with 1288 participants, covering the period from 2020 to 2024, were included. Results: Compared to the control group, TRE safely and significantly reduced body fat percentage, fat mass, lean mass, body mass, BMI, and waist circumference (MDpooled = −2.14 cm, 95% CI = −2.88~−1.40, *p* < 0.001), and increased low-density lipoprotein (LDL) (MDpooled = 2.70, 95% CI = 0.17~5.22, *p* = 0.037), but it did not alter the total cholesterol, high-density lipoprotein, and triglycerides (MDpooled = −1.09~1.20 mg/dL, 95% CI −4.31~5.47, *p* > 0.05). Subgroup analyses showed that TRE only or TRE-caloric restriction with an eating window of 6 to 8 h may be appropriate for losing body fat and overall weight. Conclusions: This work provides moderate to high evidence that TRE is a promising dietary strategy for fat loss. Although it may potentially reduce lean mass and increase LDL, these effects do not pose significant safety concerns. This trial was registered with PROSPERO as CRD42023406329.

## 1. Introduction

Obesity is a major risk factor for multiple diseases and threatens well-being, life expectancy, and the quality of life in humans [1,2,3]. Eating behaviors, which can alter energy intake, are essential in the etiology of obesity [4,5]. Changing eating behaviors in order to balance energy intake and expenditure is thus one common strategy to prevent or treat obesity [1,6].

The timing of food consumption appears to play a critical role in metabolic health [7]. In recent years, “intermittent fasting,” characterized by shifting mealtimes regardless of meal content, has received much attention for treating obesity. Time-restricted eating (TRE) is one type of intermittent fasting with the goal of improving the regulation of circadian rhythms generated by the endogenous biological clock [8]. Studies have shown that limiting daily food consumption to a 4–12 h period, during which the body is most receptive to food, may improve metabolic health [7,9]. Moreover, TRE is particularly intriguing and well-accepted by many individuals because it is the only dietary intervention that does not directly emphasize caloric restriction or require a prescribed number of fasting days [7].

Multiple studies have examined the benefits of TRE for weight loss [10,11]. However, reports of its efficacy and safety (i.e., adverse events and side effects) are inconsistent. For example, Zhang et al. [12] observed that TRE reduced adiposity but increased low-density lipoprotein cholesterol (LDL) [12]. Lowe et al. showed in another study that TRE did not reduce body fat and instead resulted in a decrease in appendicular lean mass index [13]. A previous systematic review and meta-analyses have also concluded that the impact of TRE was unclear and appears to be nuanced.

The inconsistent results of TRE intervention studies may be due, in part, to the fact that such trials have targeted very different populations and have utilized different TRE designs (e.g., eating windows length). Therefore, in this systematic review and meta-analysis, we aimed to characterize the efficacy and safety of TRE for fat loss, and related blood lipid composition, in adults with overweight and obese by including the most up-to-date peer-reviewed publications. Subgroup analyses were also completed to explicitly compare the influence of TRE intervention characteristics (e.g., eating window length, study duration, participant age, and intervention strategies) on its effectiveness [14,15,16], in the hopes of providing critical knowledge to inform the design of future research using TRE.

## 2. Methods

### 2.1. Design

A systematic review and meta-analysis of the literature were performed according to the Preferred Reporting Items for Systematic Reviews and Meta-Analyses (PRISMA) guidelines [17]. This study was registered with PROSPERO (CRD42023406329) (available online: https://www.crd.york.ac.uk/PROSPERO/display_record.php?RecordID=406329 URL (accessed on 4 July 2023)).

### 2.2. Literature Search Strategy

Seven electronic databases (PubMed, Web of Science, EMBASE, Cochrane Library, Medline, Scopus, and SPORTDiscus) were searched for publications from their inception to June 1, 2024. The search strategy was as follows: (“Time Restricted Eating” or “Eating, Time Restricted” or “Time Restricted Fasting” or “Fasting, Time Restricted,” or “Restricted Fasting, Time” or “Time-Restricted Feeding” or “Feeding, Time Restricted” or “Time Restricted Feedings” or “Time Restricted Meal” or “Time Restricted Diet” or “Early Time-Restricted Eating”) and (“Obesity” OR “Overweight” OR “Excess Weight” OR “Obese Adults” OR “Adult with Obesity” OR “Adult Obesity” OR “Obesity in Adult” OR “Obese” OR “Adiposity”). A manual search of the bibliographic references for extracted articles and existing reviews was conducted to identify studies not captured in the electronic searches. Detailed results can be found in Appendix A.

### 2.3. Inclusion and Exclusion Criteria

The inclusion criteria were as follows: (1) Participants: age ≥ 18 years old; BMI ≥ 25 kg/m^2^; no history of cardiovascular disease, metabolic disease, or gastrointestinal disease; not undergoing other fat loss measures (e.g., medications or surgery); and not pregnant or breastfeeding. (2) Intervention: the experimental group was a TRE that was used for fasting for 12 to 20 h over six weeks, while the control group had unlimited eating time. (3) Outcomes: the primary outcomes were body fat percentage, fat mass, and lean mass; the secondary outcomes were body mass index, waist circumference, and lipid profile (e.g., total cholesterol, (TC), high-density lipoprotein cholesterol (HDL), low density lipoprotein cholesterol (LDL), and triglycerides (TG)). (4) Study designs: at least one TRE intervention group and one control group were included in the study.

Studies were excluded if they were (1) animal trials, (2) written in a non-English language, (3) without specific data, (4) review and conference articles, and (5) repeated publications.

### 2.4. Data Extraction

Data extraction was carried out independently by two authors (YX and KZ), and any discrepancies in decisions were resolved through discussions with the other authors (DB and JZ) until a consensus was achieved. Extracted data included author names, year of publication, study design, sample characteristics, TRE protocol, intervention duration, outcomes, and adverse events. We extracted baseline and post-intervention data to calculate the change in outcomes when data on the change in outcomes were not directly available. The formulas [18] were as follows, where the R was set at 0.5 [19,20]:Mean (change) = Mean (final) − Mean (baseline)
$$\mathrm{SD}\;(\mathrm{change})=\sqrt{SD(baseline)²+SD(final)²-(2\times R\times SD(baseline)\times SD(final))}$$

We extracted relevant data using the semi-automated extraction tool WebPlotDigitizer (https://automeris.io/WebPlotDigitizer/, Version 4.6 URL (accessed on 2 June 2024)) for studies when the data could not be obtained by contacting the authors [21,22].

### 2.5. Quality Assessment

Two investigators (YX and KZ) independently identified the risk of bias for the relevant outcomes using the licensed Excel tool to implement the two revised Cochrane risk-of-bias tools for RCTs. The discrepancies were discussed and determined with a third investigator (DB).

### 2.6. Statistical Analysis

The weighted mean difference (MD) of the outcomes was calculated with a 95% confidence interval (CI) to evaluate the effect of the TRE on primary and secondary outcomes. When the SDs for the outcome indicators were not directly available, the SDs were calculated based on the SE or CI of the group mean according to the procedure described in the Cochrane Handbook for Systematic Review of Interventions (Chapter 6.5.2.3) [23].

A meta-analysis was performed in Stata MP 17.0 (STATA Corp., College Station, TX, USA) using the inverse variance method. Heterogeneity was assessed by measuring the inconsistency (I^2^ statistic) of intervention effects among the trials. The level of heterogeneity was interpreted according to guidelines from the Cochrane Collaboration: trivial (<25%), low (25~50%), moderate (50~75%), and high (>75%) [24]. A random effects model was employed to estimate pooled effects, anticipating heterogeneity due to variations in participants and interventions across studies. A subgroup analysis was used to analyze potential sources of heterogeneity. The publication bias was assessed by the funnel plot and Egger’s test. If significant asymmetry was detected, the trim-and-fill method was applied for sensitivity analysis [25]. Statistical significance was defined as *p* < 0.05.

Additionally, the quality of evidence for outcomes was evaluated using the Grading of Recommendations Assessment, Development, and Evaluation (GRADE), which characterizes the evidence on the study limitations, imprecision, inconsistency, indirectness, and publication bias [26,27].

## 3. Results

### 3.1. Study Selection

The screening flow diagram is shown in Figure 1. A total of 3329 relevant publications were retrieved (PubMed: *n* = 613, Web of Science: *n* = 887, Medline: *n* = 202, SPORTDiscus: *n* = 210, EMBASE: *n* = 441, Cochrane: *n* = 699, Scopus: *n* = 277, and Manual search: *n* = 0), and 3, 260 publications were excluded after reviewing the titles and abstracts. After the evaluation of the full texts, 49 of the 69 publications were removed, and thus, 20 publications consisting of 20 studies were included in the following analyses (Table 1). Uniquely, two publications [10,28] included two randomized controlled trials.

### 3.2. Quality Assessment

The overall risk of bias judgment was made according to five domain-level assessments (Figure 2). Seven studies had concerns about the overall risk of bias [11,21,29,33,36,37,41]. Four studies lacked specific information about the randomization process, and the allocation sequence’s concealment raised some concerns regarding the randomization process [11,29,37,41]. Seven studies lacked necessary information on analyzing the effect of assignment on intervention [21,29,31,32,33,36,42], raising concerns about the deviations from intended interventions. The risk of bias for missing outcome data, the measurement of the outcome, and the selection of the reported result was considered low for all studies.

### 3.3. Assessment of Evidence Certainty

We further assessed the meta-results of GRADE certainty evidence. The certainty of the evidence for the overall body composition, anthropometric measures, and blood lipid was considered “high to moderate” (Appendix A).

### 3.4. Characteristics of the Included Studies

#### 3.4.1. Participants

The included publications were conducted in different countries, including the USA [10,11,13,29,31,40,41], Norway [28], Brazil [21,33], Iran [39], Poland [30,34,38], China [12,35], and Germany [32], between 2020 and 2024. Twenty publications included 1288 participants (i.e., 936 women and 352 men) with a mean age range from 23.1 to 69.4 years and a mean BMI from 26 to 40.7 kg/m^2^. Table 1 summarizes the characteristics of all the included participants.

Participants’ occupational backgrounds included low-income women (*n* = 1) [33], non-night shift workers (*n* = 6) [10,13,28,33,36,37], and those with a standardized work schedule (*n* = 1) [29]. Body mass stabilization consisted of participants with recent stable body mass control (*n* = 9) [10,11,12,33,36,37,40,41,42]; non-low body fat high-level athletes and individuals specifically training for national and international competitions (*n* = 1) [32], and participants not taking fat loss medication or enrolled in a weight control or fat loss program (*n* = 4) [21,28,35,39]. Eight publications reported the physical activity levels of participants at the time of study participation, including physically active (*n* = 2) [28,32] and sedentary or physically inactive (*n* = 6) [10,11,31,37,41,42].

#### 3.4.2. Study Design

All the publications consisted of randomized controlled trials. Three publications [10,12,37] utilized a three-arm design of two intervention arms and a control arm. One publication [28] included four experimental groups, that is, TRE (energy intake restricted to 10 h of eating per day), HIIT (supervised treadmill exercise three times per week), combined (TRE + HIIT), and control (no intervention) (Table 1).

### 3.5. Intervention Characteristics

#### 3.5.1. Diet and Exercise Interventions

Nine publications used TRE [10,12,13,28,29,30,34,38,40] only. The other 11 publications combined TRE with other types of training and/or dietary protocols (i.e., high-intensity interval training (*n* = 2) [28,42]; concurrent exercise training (*n* = 1) [31]; caloric restriction (*n* = 5) [21,33,35,37,39]; and caloric restriction with exercise recommendation (*n* = 3)).

Participants were either allowed to eat as much as they wanted during the eating time window [10,12,13,28,29,30,34,38,40] or they were required to have similar daily amounts of energy intake and consumption between the TRE and control groups, eliminating the potential influences of the between-group variance of energy exchange on the observations [11,21,31,33,35,36,37,39,42]. In addition, one study used the Macronutrient-Based Diet (MBD) as a control group [32].

#### 3.5.2. TRE Protocol

The duration of the TRE intervention ranged from 6 to 52 weeks (12 months) (i.e., 6 (*n* = 3) [30,34,38], 7 (*n* = 1) [28], 8 (*n* = 6) [10,12,21,31,37,39], 12 (*n* = 3) [13,29,42], 14 (*n* = 3) [11,32,41], 39 (*n* = 1) [36], and 52 (*n* = 3) [33,35,40] weeks). The eating window length of each TRE was between 4 and 12 h (i.e., 4 (*n* = 1) [10], 6 (*n* = 2) [10,12], 8 (*n* = 13) [11,13,21,29,30,31,32,34,35,37,38,41,42], 10 (*n* = 3) [28,36,39], and 12 (*n* = 1) [33] hours). In addition, one publication [40] used an 8-h eating window to reduce the body mass of participants during the first 6-month period, and a 10-h eating window to maintain the body mass during the second 6-month period.

The timing of the eating plan on each day was different across the publications. Only two publications [28,36] provided recommendations on when to start or end eating, thirteen publications recommended the timing of eating, and the other two [29,33] did not. Specifically, in those thirteen publications with designed timing, nine [10,13,30,31,32,34,38,39,40] used delayed time-restricted eating (dTRE; participants ate between 12:00 and 20:00) and four [11,35,41,42] used early time-restricted eating (eTRE; participants ate between 7:00 and 16:00). In addition, two publications [12,37] compared the effects of eTRE and dTRE interventions, and another study allowed participants to choose the protocol of either eTRE or dTRE [21] (Table 1).

#### 3.5.3. Outcome Measurement

The outcomes of body composition included fat mass (*n* = 18) [10,11,12,13,21,28,29,30,31,32,34,35,36,37,39,40,41,42], lean mass (*n* = 16) [10,11,12,13,21,29,30,31,32,34,35,36,37,40,41,42], and body fat percentage (*n* = 11) [12,13,21,29,30,33,34,35,37,38,39], with the addition of body weight estimates based on the total fat mass and body fat percentage (*n* = 5) [28,31,36,42], as calculated by a formula detailed in Appendix A. The methods of measuring body composition include dual-energy X-ray absorptiometry (*n* = 10) [10,11,13,29,31,35,36,37,40,41] and bioelectrical impedance measurements (*n* = 10) [12,21,28,30,32,33,34,38,39,42].

The outcomes of anthropometric measures included body mass (*n* = 19) [11,12,13,21,28,29,30,31,32,33,34,35,36,37,38,39,40,41,42], BMI (*n* = 13) [12,13,30,31,32,33,34,35,37,38,39,40,42], and waist circumference (*n* = 13) [11,12,13,21,31,32,33,34,35,38,39,40,42]. Waist circumference was measured using three methods: at the midpoint between the top of the iliac crest and the base of the lowest ribs (*n* = 5) [11,13,33,39,42], at the location of the minimum abdominal circumference (*n* = 2) [21,32] or two finger widths above the umbilicus (*n* = 1) [31].

The outcomes of the blood sample composition included TC (*n* = 10) [11,12,13,21,28,31,35,37,40,42]; HDL (*n* = 11) [10,11,12,13,28,29,31,35,37,40,42]; LDL (*n* = 10) [10,11,12,13,28,29,35,37,40,42]; and TG (*n* = 10) [10,11,12,13,28,29,35,37,40,42]. Additionally, most of these publications also examined the effects of TRE on visceral and subcutaneous fat, water, and resting metabolic rate.

### 3.6. Adverse Events and Side Effects

Thirteen publications [12,13,21,28,29,30,32,34,36,38,39,40,42] did not report information on the reported side effects or adverse events. Seven publications [10,11,31,35,37,40,41] did report the occurrence of adverse events, and several side effects or adverse events were reported, including headaches, nausea, diarrhea, hunger, fatigue, dizziness, decreased appetite, epigastric pain, indigestion, constipation, vertigo, gastric reflux, or stomachache. All reported events were mild, transient, and occurred with a similar frequency across the intervention and control groups.

## 4. Meta-Analysis

Based on the heterogeneity and the number of studies that were available for grouping, we performed subgroup analyses by comparing the eating window length (i.e., 6 to 8 h and 10 to 12 h), study duration (i.e., ≤39 weeks and ≥39 weeks), participant average age (i.e., 23 to 45-year-old young adults, 45 to 65-year-old middle-aged adults, and ≥65 year-old older adults), and TRE intervention strategies (i.e., TRE-only, TRE combined with CR, TRE combined with exercise, and TRE combined with CR and exercise recommendations) (Appendix A). Since the study by Cienfuegos et al. [10] used a 4-h eating time and the study by Isenmann et al. [32] directly compared TRE to MBD, these studies were not included in the meta-analysis.

### 4.1. Effects of TRE on Body Composition

#### 4.1.1. Body Fat Percentage

Five studies [12,21,30,33,34] showed that TRE decreased the body fat percentage compared to control, while another six studies [13,29,35,37,38,39] reported that TRE did not alter this outcome (Table 2). Pooled data indicate that TRE decreased the body fat percentage [Fixed: MD_pooled_ = −0.48%, 95% CI (0.83, −0.13), *p* = 0.007, Figure 3A] with trivial heterogeneity (*I*^2^ = 0%, *p* = 0.522). The funnel plot and Egger’s test (t = −1.61, *p* = 0.129) indicated no publication bias.

#### 4.1.2. Fat Mass

Twelve studies [10,12,21,28,30,31,34,39,40,41,42] showed that TRE reduced fat mass compared to the control intervention, while another six studies [11,13,29,35,36,37] reported no such effect on this outcome (Table 2). Pooled data indicated that TRE reduced fat mass [Random: MD_pooled_ = −1.40 kg, 95% CI (−1.94, −0.85), *p* < 0.001, Figure 3B] with trivial heterogeneity *(I*^2^ = 49.4%, *p* = 0.009). The funnel plot and Egger’s test (t = 0.35, *p* = 0.731) indicated no publication bias.

Subgroup analyses revealed that (1) TRE with an eating window of 6 to 8 h reduced fat mass (MD = −1.66 kg, 95% CI (−2.31, −1.01), *p* < 0.001), while that with an eating window of 10 to 12 h did not (MD = −0.65 kg, 95% CI (−1.39, 0.90), *p* = 0.085); (2) the TRE of both ≤ 39 weeks (MD = −1.36 kg, 95% CI (−2.00, −0.72) *p* < 0.001) and ≥39 weeks study durations reduced fat mass (MD = −1.51 kg, 95% CI (−2.51, −0.51), *p* = 0.003); (3) the reduction in fat mass was observed in both young adults (MD = −1.61 kg, 95% CI (−2.23, −0.99), *p* < 0.001) and middle-aged adults (MD = −0.64 kg, 95% CI (−1.27, −0.01), *p* = 0.045), but not in older adults (MD = −1.80 kg, 95% CI (−4.66, 1.06), *p* = 0.218); and (4) a decrease in fat mass was observed in the TRE-only group (MD = −1.26 kg, 95% CI (−1.89, −0.62), *p* < 0.001), TRE combined with CR (MD = −0.83 kg, 95% CI (−1.53, −0.12), *p* = 0.022), TRE combined with exercise (MD = −3.07 kg, 95% CI (−6.02, −0.12), *p* = 0.041), and TRE combined with CR and exercise recommendations (MD = −1.87 kg, 95% CI (−3.17, −0.57), *p* = 0.005, Appendix A).

#### 4.1.3. Lean Mass

Four studies [10,12,21,29] reported that TRE reduced lean mass compared to the control intervention, while eleven studies [11,13,30,31,34,35,36,37,40,41,42] did not report a change in this outcome (Table 2). Analyses of pooled data revealed that TRE reduced lean mass [Random: MD_pooled_ = −0.64 kg, 95% CI (−0.87, −0.40), *p* = < 0.001, Figure 3C] with trivial heterogeneity (*I*^2^ = 12.6%, *p* = 0.314). The funnel plot and Egger’s test (t = 0.83, *p* = 0.421) indicated no publication bias.

Subgroup analyses demonstrated that (1) TRE with an eating window of 6 to 8 h reduced lean mass (MD = −0.74 kg, 95% CI (−0.95, −0.53), *p* < 0.001), while eating windows of 10 to 12 h did not (MD = −0.01 kg, 95% CI (−0.60, 0.58), *p* = 0.968); (2) a TRE duration of ≤ 39 weeks decreased lean mass (MD = −0.73 kg, 95% CI (−0.95, −0.51) *p* < 0.001), while ≥ 39 weeks did not (MD = −0.25 kg, 95% CI (−0.84, 0.35), *p* = 0.416); 3) a reduction in lean mass was observed in both young adults (MD = −0.53 kg, 95% CI (−0.77, −0.28), *p* < 0.001) and middle-aged adults (MD = −1.09 kg, 95% CI (−1.51, −0.66), *p* < 0.001), but not in older adults (MD = 0.10 kg, 95% CI (−2.45, 2.66), *p* = 0.936); and 4) a decrease in lean mass was observed in the TRE-only group (MD = −1.04 kg, 95% CI (−1.36, −0.72), *p* < 0.001), while TRE combined with CR (MD = −0.30 kg, 95% CI (−0.85, 0.26), *p*= 0.293), TRE combined with exercise (MD = −0.01 kg, 95% CI (−1.78, 1.76), *p* = 0.989), and TRE combined with CR and exercise recommendations did not (MD = −0.46 kg, 95% CI (−0.75, −0.17), *p* = 0.002, Appendix A).

### 4.2. Effects of TRE on Anthropometric Measures

#### 4.2.1. Body Mass

Twelve studies [11,12,21,28,29,31,34,39,40,41,42] reported that TRE reduced body mass compared to control, while another seven studies [13,30,33,35,36,37,38] did not demonstrate this effect (Table 2). An analysis of the pooled data indicated that TRE had no effect on body mass [Random: MD_pooled_ = −2.11 kg, 95% CI (−2.89, −1.33), *p* < 0.001, Figure 4A] with moderate heterogeneity (*I*^2^ = 62.0%, *p* = 0.000). The funnel plot and Egger’s test (t = 1.03, *p* = 0.318) indicated no publication bias.

Subgroup analyses showed that 1) TRE with an eating window of 6 to 8 h reduced body mass (MD = −2.73 kg, 95% CI (3.42, −2.05), *p* < 0.001), while those with an eating window of 10 to 12 h did not (MD = −0.67 kg, 95% CI (−1.44, −0.11), *p* = 0.091); 2) the TRE of both ≤ 39 weeks (MD = −2.13 kg, 95% CI (−3.01, −1.25) *p* < 0.001) and ≥ 39 weeks (MD = −2.02 kg, 95% CI (−3.94, −0.09), *p* = 0.007) reduced body mass; 3) a reduction in body mass was observed in young adults (MD = −2.24 kg, 95% CI (−3.13, −1.36), *p* < 0.001), but not in middle-aged adults (MD = −1.13 kg, 95% CI (−2.31, 0.04), *p* = 0.058) or older adults (MD = −1.79 kg, 95% CI (−4.87, 1.28), *p* = 0.252); and 4) the TRE-only group (MD = −2.39 kg, 95% CI (−3.42, −1.35), *p* < 0.001), TRE combined with CR (MD = −1.22 kg, 95% CI (−2.13, −0.30), *p* = 0.009), and TRE combined with CR and exercise recommendations (MD = 2.48 kg, 95% CI (−4.15, −0.81), *p* = 0.004) reduced body mass, while TRE combined with exercise did not (MD = −3.76 kg, 95% CI (−7.52, 0.01), *p* = 0.051, Appendix A).

#### 4.2.2. Body Mass Index

Seven studies [12,30,31,34,39,40,42] reported that TRE decreased BMI compared to the control intervention, while another five studies [13,33,35,37,38] did not observe a change in this outcome (Table 2). An analysis of the pooled data indicated that TRE had no effect on BMI [Random: MD_pooled_ = −0.75 kg/m^2^, 95% CI (−1.12, −0.38), *p* < 0.001, Figure 4B] with moderate heterogeneity (*I*^2^ = 66.0%, *p* = 0.000). The funnel plot and Egger’s test (t = −0.80, *p* = 0.444) indicated no publication bias.

Subgroup analyses revealed that (1) TRE with an eating window of 6 to 8 h decreased BMI (MD = −0.98 kg/m^2^, 95% CI (−1.44, −0.52), *p* < 0.001), while those with an eating window of 10 to 12 h did not (MD = −0.27 kg/m^2^, 95% CI (−0.58, 0.05) *p* = 0.097); (2) a TRE duration of both ≤ 39 weeks (MD = −0.70 kg, 95% CI (−1.13, −0.28) *p* = 0.001) and ≥ 39 weeks (MD = −0.98 kg/m^2^, 95% CI (−1.85, −0.11), *p* = 0.027) decreased BMI; (3) a reduction in BMI was observed in both young adults (MD = −0.86 kg/m^2^, 95% CI (−1.37, −0.34), *p* < 0.001), and older adults (MD = −0.82 kg/m^2^, 95% CI (−1.54, −0.11), *p* = 0.025), but not in middle-aged adults (MD = −0.34 kg/m^2^, 95% CI (−0.68, 0.00), *p* = 0.051); and (4) the TRE-only group (MD = −0.89 kg/m^2^, 95% CI (−1.36, −0.42), *p* < 0.001) and TRE combined with CR (MD = −0.31 kg/m^2^, 95% CI (−0.60, −0.57), *p* = 0.037) decreased BMI, while TRE combined with exercise did not (MD = −2.47 kg/m^2^, 95% CI (−5.51, 0.57), *p* = 0.111, Appendix A).

#### 4.2.3. Waist Circumference

Seven studies [12,21,30,34,39,40,42] reported that TRE reduced waist circumference compared to the control group, while another five publications [11,13,31,33,35] reported no effect on this outcome (Table 2). An analysis of the pooled data indicated that TRE reduced waist circumference [Random: MD_pooled_ = −2.14 cm, 95% CI (−2.88, −1.40), *p* < 0.001, Figure 4C] with low heterogeneity (*I*^2^ = 0%, *p* = 0.711). The funnel plot and Egger’s test (t = −1.03, *p* = 0.328) indicated no publication bias.

### 4.3. Effects of TRE on Blood Lipid

Ten studies [11,13,28,29,35,37,40,42,43] showed that TRE cannot significantly change LDL compared to the control, and only one [12] showed that TRE can significantly increase LDL (Table 2). An analysis of the pooled data indicated that TRE increased LDL [Random: MD_pooled_ = 2.70 mg/dL, 95% CI (0.17, 5.22), *p* = 0.037, Figure 5C] with low heterogeneity (I^2^ = 0%, *p* = 0.573). The funnel plot and Egger’s test (t = −1.24, *p* = 0.246) indicated no publication bias. For other outcomes of blood lipid, all the studies showed that TER cannot induce significant effects on TC (*n* = 11, Random: MD_pooled_ = 1.20 mg/dL, 95% CI (−3.07, 5.47), *p* = 0.582 Figure 5A), HDL (*n* = 12, Random: MD_pooled_ = 0.53 mg/dL, 95% CI (−0.60, 1.65), *p* = 0.357, Figure 5B), or TG [*n* = 11, Random: MD_pooled_ = −1.09 mg/dL, 95% CI (−4.31, 2.13), *p* = 0.506, Figure 5D] without heterogeneity (I^2^ = 18.6~38.2%, *p* = 0.095~0.266). The funnel plots and Egger’s test indicated (t = −1.22~0.4, *p* = 0.154~0.694) no publication bias.

There were no significant differences in TC, HDL, and TG between the subgroups, except for HDL, which was significantly increased in TRE combined with CR (MD = 2.38 mg/dL, 95% CI (0.17, 4.60), *p* = 0.035), and for TG, which was significantly decreased in TRE with an eating window of 6 to 8 h (MD = −4.68 mg/dL, 95% CI (−9.01, −0.36), *p* = 0.034, Appendix A).

## 5. Discussion

This systematic review and its meta-analysis provide evidence of moderate to high quality that TRE appears safe and effectively reduces body fat, yet also reduces lean mass and does not appear to affect blood lipid levels. Subgroup analyses further indicated that TRE was particularly beneficial for younger and middle-aged adults, and such effect sizes of benefits are pertaining to the design of intervention protocols (e.g., the benefits would be maximized when using a 6 to 8 h eating window length). These results thus provide important knowledge for the appropriate design of TRE protocols, helping the development of weight loss programs using TRE.

Our results indicate that implementing TRE is an effective strategy for body weight, body mass index, and waist circumference in adults with overweight and obese, via a TRE-induced reduction in body fat percentage. These benefits of TRE for fat loss may stem from physiological changes triggered by time-restricted eating, such as synchronization with endogenous circadian rhythms in both central and peripheral biological clocks [44,45]. TRE, as a form of a time-restricted diet aligned with individual circadian rhythms, modulates the eating-fasting cycle to optimize metabolic processes [9,46]. For instance, studies have shown that TRE can increase adiponectin levels during fasting, induce a moderate rise in ketone bodies, and activate AMPK (adenosine 5′-monophosphate (AMP)-activated protein kinase), which enhances insulin sensitivity, stimulates lipolysis, and promotes the secretion of lipocalin during fasting, thereby aiding in fat reduction [44,47,48]. Additionally, TRE has been observed to reduce caloric intake by about 20% to 30% due to the decreased eating frequency, further contributing to body weight loss [10,12,28,44]. It is observed that TRE with eating windows of 6 to 8 h, either alone or combined with caloric restriction, may be most effective for reducing body fat and overall body mass. Both shorter (e.g., 6 to 14 weeks) and longer (e.g., 39 to 52 weeks) durations of TRE resulted in fat loss. Combining TRE with exercise led to a significant decrease in body fat, but no change in overall body mass, likely due to the retention of lean mass, which will be further discussed in the next section. Another reason could be the “ceiling effect,” where the inherent ability of exercise therapy (e.g., HIIT) to reduce body weight [49] limits the additional benefits of TRE. Only a few studies (*n* = 3) have explored the effects of TRE combined with exercise, with differing outcomes, indicating the need for more research to fully understand these interactions. Remarkably, no significant weight loss benefits of TRE were observed in older cohorts, possibly due to insufficient statistical power from the small number of studies (*n* = 3) with limited sample sizes. It is thus worthwhile to perform studies with the focus on older adults and explore the effects of TRE on body weight and composition, as well as its potential benefits for sarcopenic obesity [50] in this population.

Interestingly, we observed that TRE reduced lean mass, which also contributed to the observed weight loss induced by TRE. Such a loss of lean mass may be attributed primarily to inadequate protein intake and changes in protein turnover (decreased muscle protein synthesis, increased catabolism, or both) [51,52,53]. Furthermore, a loss of body water may contribute, as weight loss often reduces circulation and body fluids in adults with overweight and obese [12,54]. However, this does not appear to raise safety concerns [55], as studies indicate that in adults with obese and overweight, lean mass typically accounts for only 20 to 30% of the total weight lost during weight reduction. The observed lean mass loss with TRE in this study was relatively small (about 17.1%), suggesting it is within a reasonable range [12,13,52]. Significant lean mass reduction was observed with shorter TRE durations, suggesting that muscle protein loss may occur early in the process and could decrease as ketogenesis increases [56]. Only one study, Lowe et al. [13], observed that 65% of the weight loss during a 12-week, 8-h restricted eating on TRE was lean mass. While the exact cause remains unclear, such a high percentage of lean mass loss has been linked to weight rebound and an increased risk of sarcopenia, particularly when the lean mass loss exceeds fat loss [52,57,58]. Given the limited attention to this issue [28,31,47], we still recommend combining TRE with strength training and appropriate protein supplementation to prevent lean mass loss, despite the absence of serious adverse events or sarcopenia reports [59]. Future studies are warranted to explicitly characterize the effects of TRE on muscle mass and hydration, and the related long-term influences on sarcopenia risk, providing more comprehensive knowledge to optimize TRE protocols for weight reduction without compromising health.

No significant effects of TRE on blood lipids were observed, except for an unexpected increase in LDL [12]. This increase could represent a short-term adverse reaction to dietary changes, possibly attributed to prolonged fasting and the resulting increased reliance on fat oxidation during the TRE intervention [11,12,60]. In addition, the effect of TRE on most blood lipids is negligible because we mainly focused on adults with overweight and obese without metabolic diseases, and the results of this study were mainly influenced by those observed in younger adults, such as in Zhang et al. [12], who observed that blood lipid levels in young adults with overweight (23 ± 1 years) were all within the normal range (TC 164~178, HDL 40~46, LDL 96~105, TG 84~93 mg/dL). Therefore, this suggests that the regulatory effect of TRE on lipid metabolism in this group of people is minimal and is unrelated to any safety concerns. Remarkably, TG levels were significantly lower with a 6 to 8-h eating window diet, likely due to the extended fasting period that accelerates the body’s transition into a fat-metabolizing state. During this fasting period, insulin levels decrease, which enhances lipolysis and consequently reduces TG levels [61,62,63]. However, one study demonstrated that over a 12-month period, an 8-hour time-restricted eating regimen combined with daily caloric restriction provided no additional benefits compared to daily caloric restriction alone [35]. Despite this, the present study observed an association between TRE combined with CR and an increase in HDL, which might be due to the reduction in body fat and overall weight [64]. Overall, the potential of combining TRE with CR remains promising, though further research is required to confirm its benefits. It is important to note that blood lipid metabolism is regulated by circadian rhythms, which differ among individuals [65]. To optimize the effects of TRE on blood lipids, it may be necessary to tailor the TRE protocol to align with individual circadian rhythms, potentially improving cardiovascular health [62,65].

Some limitations should be noted when interpreting the observations in this work. First, a relatively small number (*n* = 20) of studies were included in the meta-analysis and the study protocols varied. For instance, the duration of TRE interventions ranged from 6 to 52 weeks, which may potentially limit the strength of the evidence. Second, this analysis focused primarily on adults with overweight and obese without metabolic abnormalities, and most of the included studies concentrated on young and middle-aged women (i.e., 72.8% of the included participants). As a result, the findings may not be generalizable to patients with metabolic disorders such as cardiovascular disease, diabetes, or polycystic ovary syndrome. Further studies are necessary to evaluate the effectiveness of TRE interventions in other populations, such as in older men and individuals with metabolic diseases, to establish the generalizability of TRE to public health. Third, due to the limited number of publications, several sub-group analyses cannot be performed (e.g., the level of baseline physical activity in participants, which may contribute to the effects of TRE) [66]_._ Additionally, future studies should more explicitly evaluate and track side effects and adverse events, while also considering the quantification of protein intake and the incorporation of regular physical exercise.

This work provides moderate to high amounts of evidence that TRE is a promising dietary strategy for fat loss. Although it may potentially reduce lean mass and increase LDL, these effects do not pose significant safety concerns.

## Figures and Tables

**Figure 1 nutrients-16-03390-f001:**
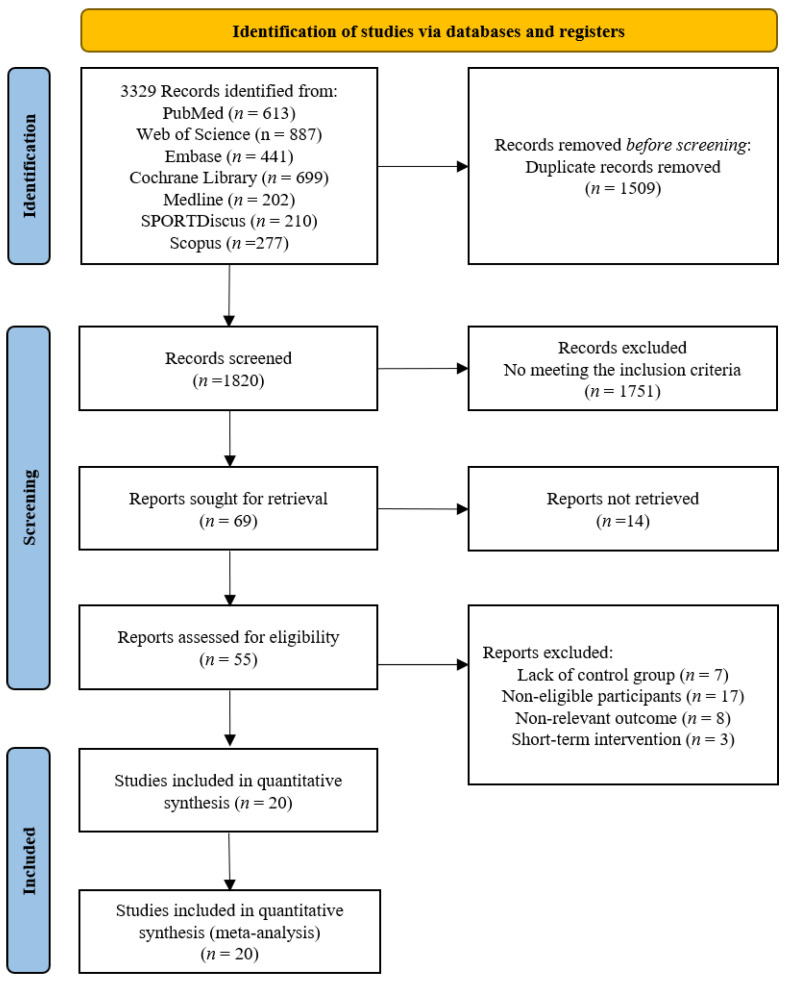
Flow chart of the publication screening.

**Figure 2 nutrients-16-03390-f002:**
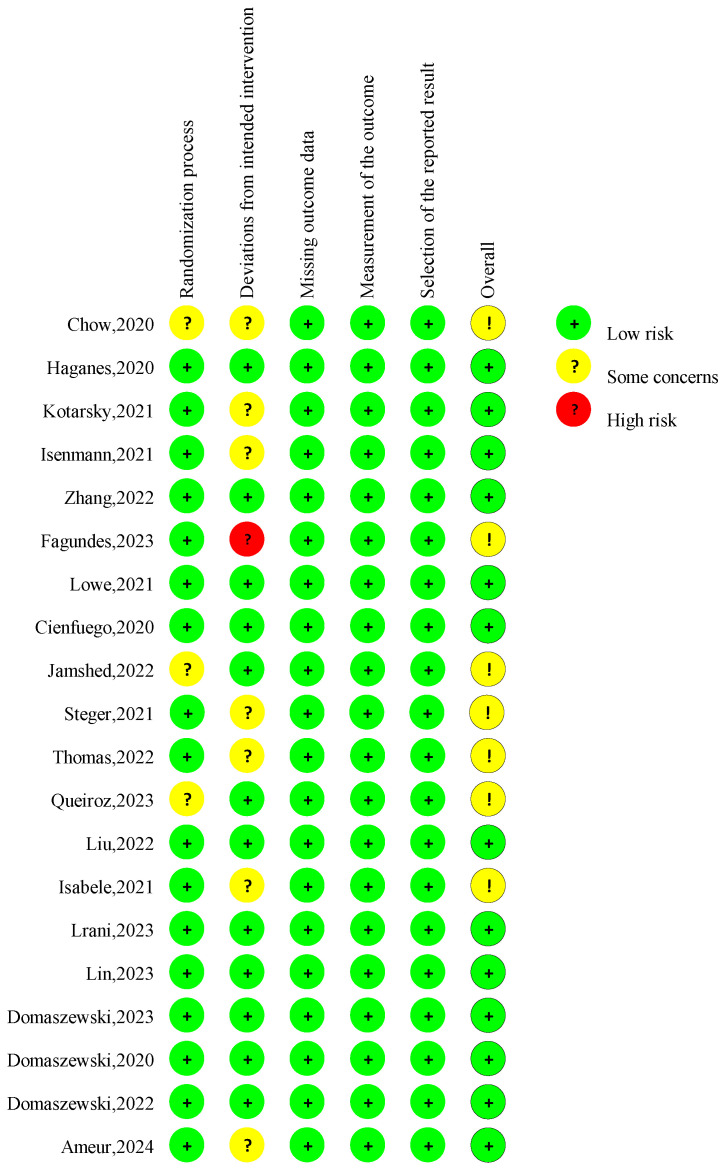
Risk of bias assessment in the RCTs in the included studies. A total of 20 publications [10,11,12,13,21,28,29,30,31,32,33,34,35,36,37,38,39,40,41,42] were incorporated into the review.

**Figure 3 nutrients-16-03390-f003:**
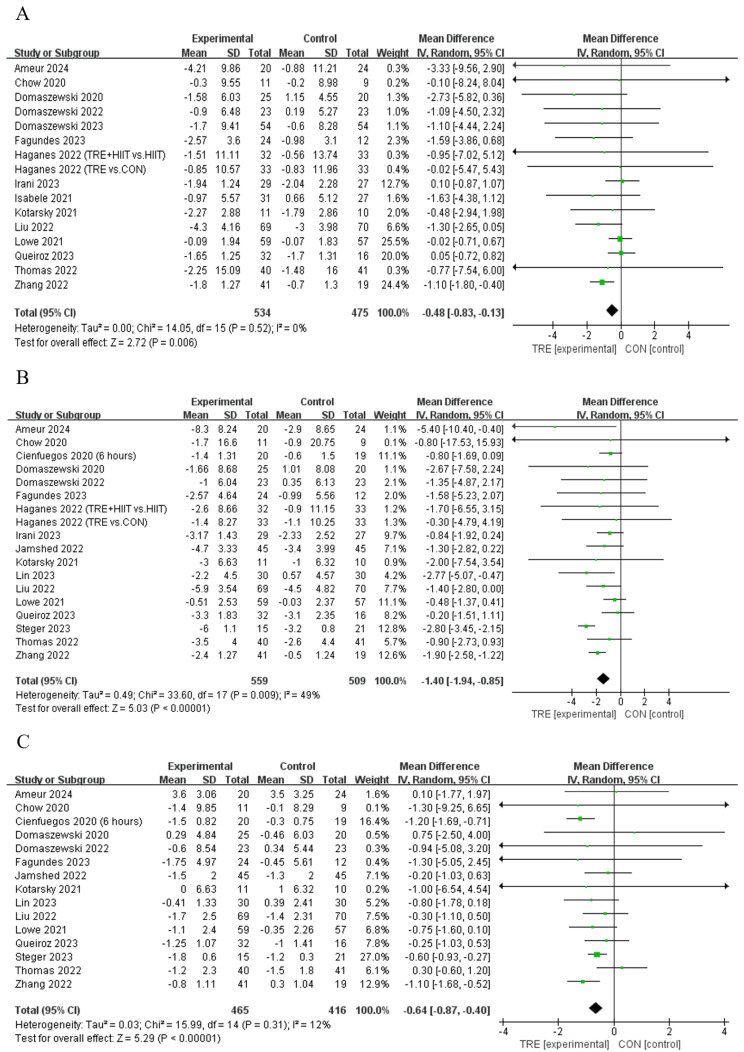
Meta−analysis results of (**A**) body fat percentage [12,13,21,28,29,30,31,33,34,35,36,37,38,39,42], (**B**) fat mass [10,11,12,13,21,28,29,30,31,34,35,36,37,39,40,41,42], and (**C**) lean mass [10,11,12,13,21,29,30,31,34,35,36,37,40,41,42].

**Figure 4 nutrients-16-03390-f004:**
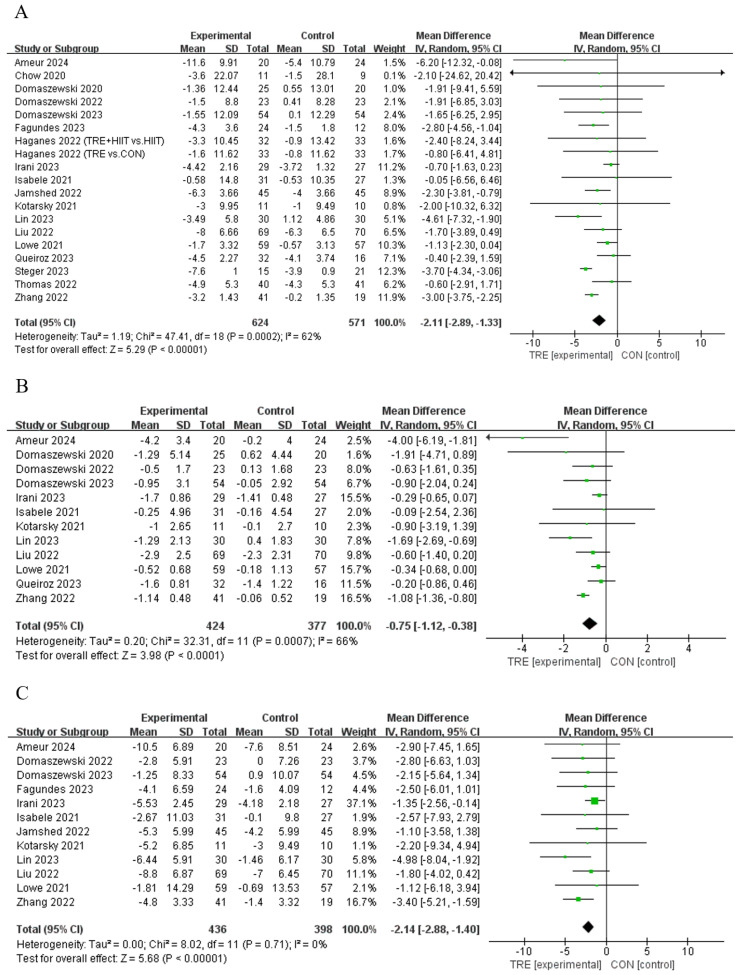
Meta−analysis results for (**A**) body mass [11,12,13,21,28,29,30,31,33,34,35,36,37,38,39,40,41,42], (**B**) body mass index [12,13,30,31,33,34,35,37,38,39,40,42], and (**C**) waist circumference [11,12,13,21,31,33,34,35,38,39,40,42].

**Figure 5 nutrients-16-03390-f005:**
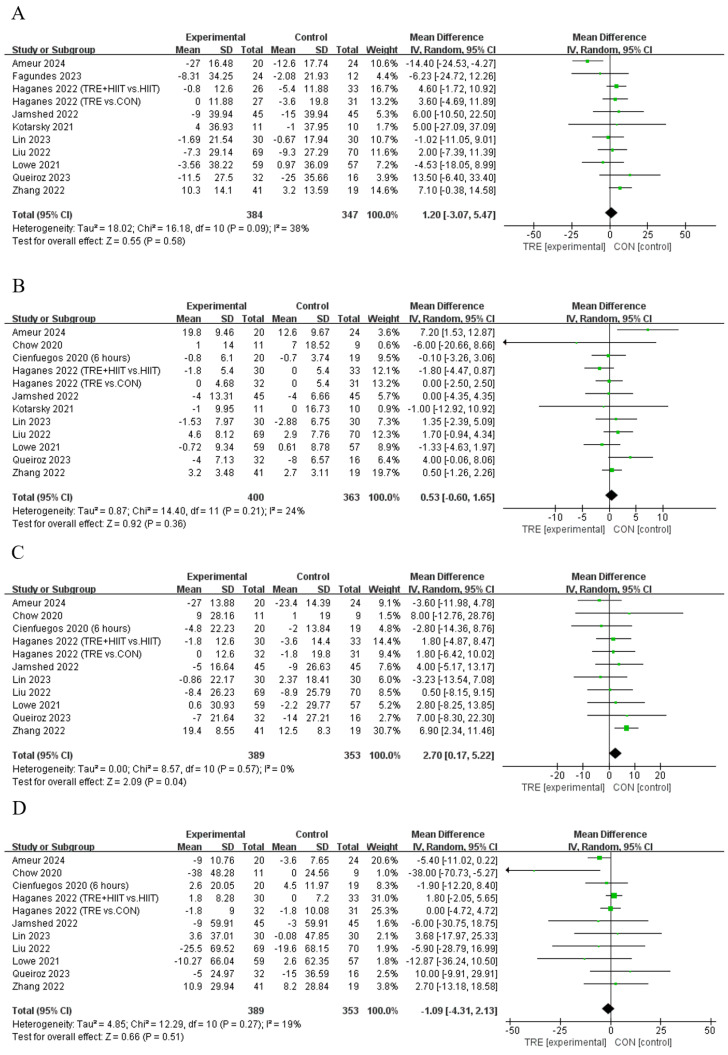
Meta−analysis results for (**A**) total cholesterol [11,12,13,21,28,31,35,37,40,42], (**B**) high−density lipoprotein cholesterol [10,11,12,13,28,29,31,35,37,40,42], (**C**) low−density lipoprotein cholesterol [10,11,12,13,28,29,35,37,40,42], and (**D**) triglycerides [10,11,12,13,28,29,35,37,40,42].

**Table 1 nutrients-16-03390-t001:** Characteristics of the included publications (20).

Study	Participants	BMI (kg/m^2^)	Group (*n*)	TRE Protocol (Fasting: Eating)	TRE Eating Time	Diet and Exercise Interventions	Duration (Weeks)
Chow et al., 2020 [29]	3 M, 17 F; 46 ± 12 y; overweight and obese adults	34 ± 8	TRE (11) CON (9)	16:08	Self-selecting		12
Cienfuegos et al., 2020 [10]	5 M, 53 F; 46 ± 3 y; obese adults	37 ± 1	4-h TRE (19) 6-h TRE (20) CON (19)	20:04; 18:06	15:00–19:00; 13:00–19:00		8
Domaszewski et al., 2020 [30]	45 F; 65 ± 5 y; overweight adults	28 ± 5	TRE (25) CON (20)	16:08	12:00–20:00		6
Lowe et al., 2020 [13]	70 M, 46 F; 47 ± 11 y; overweight and obese adults	31 ± 4	TRE (59) CON (57)	16:08	12:00–20:00		12
Kotarsky et al., 2021 [31]	3 M, 18 F; 44 ± 7 y; overweight and obese adults	30 ± 3	TRE (11) CON (10)	16:08	12:00–20:00	Concurrent training: aerobic and supervised resistance training	8
Isenmann et al., 2021 [32]	14 M, 21 F; 28 ± 5 y; overweight adults	26	TRE (18) MBD (17)	16:08	12:00–20:00	At least two training sessions per week at the local gym; diets contain 45–65% carbohydrate, 20–35% fat, and 20–35% protein	14
Isabele et al., 2021 [33]	58 F; 31.4 ± 7 y; overweight and obese females	33 ± 4	HD + TRE (31) HD (27)	12:12	Self-selecting	Subtract 500–1000 kcal from total energy consumption	52
Haganes et al., 2022 [28]	131 F; 36 ± 6 y; overweight and obese females	32 ± 4	TRE + HIIT (33) CON (32) TRE (33) HIIT (33)	14:10	No later than 20:00	High-intensity interval training supervised treadmill training three times per week	7
Domaszewski et al., 2022 [34]	46 M; 70 ± 3 y; overweight adults	28 ± 2	TRE (23) CON (23)	16:08	12:00–20:00		6
Liu et al., 2022 [35]	71 M, 68 F; 32 ± 9 y; overweight and obese adults	32 ± 3	TRE (69) CON (70)	16:08	8:00–16:00	Diets contain 40–55% carbohydrate, 5–20% protein, and 20–30% fat; the regimen accounts for approximately 75% of the subject’s baseline daily caloric intake (F:1200–1500 kcal/d) (M:1500–1800 kcal/d)	52
Zhang et al., 2022 [12]	33 M, 27 F; 23 ± 1 y; overweight and obese adults	28 ± 1	eTRE (41) CON (19)	18:06	7:00–13:00; 12:00–18:00		8
Jamshed et al., 2022 [11]	18 M, 72 F; 43 ± 11 y; obese adults	40 ± 7	eTRE + ER (45) CON + ER (45)	16:08	7:00–15:00	Low-calorie diet (500 kcal/d lower than resting energy expenditure) and 75–150 min/week of exercise is recommended	14
Thomas et al., 2022 [36]	12 M, 69 F; 38 ± 8 y; overweight and obese adults	34 ± 6	eTRE + DCR (41) DCR (40)	14:10	Within 3 h of waking	Diets contain 35% caloric restriction (10% reduction in personal resting energy expenditure); 150 min/week of moderate-intensity physical activity is recommended	39
Queiroz et al., 2023 [37]	6 M, 42 F; 30 ± 6 y; overweight and obese adults	31 ± 3	eTRE (16) dTRE (16) CON (16)	16:08	8:00–16:00; 12:00–20:00	Energy restriction (resting energy*1.4 PAL-25% of daily energy requirement), a diet consisting of 50% carbohydrate, 20% protein, and 30% fat	8
Fagundes et al., 2023 [21]	36 F; 35 ± 9 y; overweight and obese females	30 ± 3	TRE (24) CON (12)	16:08	8:00–16:00 or 12:00–20:00	Diets contain 40–45% carbohydrate, 30–35% fat, and 20–25% protein; energy limitation range 513–770 kcal/d	8
Domaszewski et al., 2023 [38]	51 M, 57 F; 69 ± 4 y; overweight adults	28 ± 3	TRE (54) CON (54)	16:08	12:00–20:00		6
Irani et al., 2023 [39]	56 F; 42 ± 9 y; overweight and obese females	31 ± 4	TRE (29) CON (27)	14:10	10:00–20:00	Low-calorie diet (total calories calculated by multiplying the Mifflin-St. Joer formula by the coefficient of physical activity and the thermic effect of the food, from which 300–500 kcal are subtracted), with 52% carbohydrates, 18% proteins, and 30% fats	8
Lin et al., 2023 [40]	10 M, 50 F; 40 ± 11 y; obese adults	38 ± 6	TRE (30) CON (30)	16:08, 14:10	BM loss phase (12:00–20:00); BM maintenance phase (10:00–20:00)		52
Steger et al., 2023 [41]	10 M, 26 F; 44 ± 12 y; obese adults	38 ± 6	eTRE + ER (15) CON + ER (21)	16:08	7:00–15:00	Low-calorie diet (500 kcal/d lower than resting energy expenditure) and 75–150 min/week of exercise is recommended	14
Ameur et al., 2024 [42]	44 F; 32 ± 10 y; obese females	35 ± 4	TRE (20) CON (24)	16:08	8:00–16:00	HIFT was performed 3 days per week on Monday, Wednesday, and Friday in the evening at the fasting window (i.e., 5:00 pm)	12

Note: Y: year; M: male; F: female; eTRE: early time−restricted eating; dTRE: delayed time−restricted eating; HD: hypoenergetic diet; HIIT: high−intensity interval training; ER: energy restriction; DCR: daily caloric restriction; PAL: physical activity level; kcal/d: kilocalorie/day; and min: minute.

**Table 2 nutrients-16-03390-t002:** Between-group comparisons of study outcomes (20).

Study	Outcomes (TRE vs. Control)									
	BF%	FM	LM	BM	BMI	WC	TC	HDL	LDL	TG
Chow et al., 2020 [29]	↔	↔	↓	↓				↔	↔	↔
Cienfuegos et al., 2020 (6 h) [10]		↓	↓					↔	↔	↔
Domaszewski et al., 2020 [30]	↓	↓	↔	↔	↓	↓				
Lowe et al., 2020 [13]	↔	↔	↔	↔	↔	↔	↔	↔	↔	↔
Kotarsky et al., 2021 [31]		↓	↔	↓	↓	↔	↔	↔		
Isabele et al., 2021 [33]	↓			↔	↔	↔				
Haganes et al., 2022 (TRE vs. CON) [28]		↓		↓			↔	↔	↔	↔
Haganes et al., 2022 (TRE + HIIT vs. HIIT) [28]		↓		↓			↔	↔	↔	↔
Domaszewski et al., 2022 [34]	↓	↓	↔	↓	↓	↓				
Zhang et al., 2022 [12]	↓	↓	↓	↓	↓	↓	↔	↔	↑	↔
Liu et al., 2022 [35]	↔	↔	↔	↔	↔	↔	↔	↔	↔	↔
Jamshed et al., 2022 [11]		↔	↔	↓		↔	↔	↔	↔	↔
Thomas et al., 2022 [36]		↔	↔	↔						
Queiroz et al., 2023 [37]	↔	↔	↔	↔	↔		↔	↔	↔	↔
Fagundes et al., 2023 [21]	↓	↓	↓	↓		↓	↔			
Domaszewski et al., 2023 [38]	↔			↔	↔					
Irani et al., 2023 [39]	↔	↓		↓	↓	↓				
Lin et al., 2023 [40]		↓	↔	↓	↓	↓	↔	↔	↔	↔
Steger et al., 2023 [41]		↓	↔	↓						
Ameur et al., 2024 [42]		↓	↔	↓	↓	↓	↓	↔	↔	↓

Note: FM: fat mass; BF%: body fat percentage; LM: lean mass; BM: body mass; WC: waist circumference; TC: total cholesterol; HDL: high−density lipoprotein cholesterol; LDL: low−density lipoprotein cholesterol; TG: triglycerides; ↓, TRE significantly (*p* < 0.05) reduced the outcome compared to the control group; ↑, TRE significantly (*p* < 0.05) improved the outcome compared to the control group; and ↔, no significant difference (*p* > 0.05) between the TRE and control group.

## Data Availability

The database supporting the conclusions of this systematic review will be available from the corresponding author upon request, pending application and approval.

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
