# Peer review of "The Effects of Time-Restricted Eating on Fat Loss in Adults with Overweight and Obese Depend upon the Eating Window and Intervention Strategies: A Systematic Review and Meta-Analysis"

_nutrients, 2024, doi:10.3390/nu16193390_

Round 1

Reviewer 1 Report

Comments and Suggestions for Authors

The systematic review and meta-analysis titled “nutrients-3230382_The effects of time-restricted eating on fat loss in adults with overweight and obese depend upon the eating window and intervention strategies: A systematic review and meta-analysis  “ is submitted to the “Nutrition and Obesity” section of the journal.

This systematic review and meta-analysis is registered with PROSPERO and follows the current guidelines, in accordance with the Preferred Reporting Items for Systematic Reviews and Meta-Analyses (PRISMA) recommendations.

The title accurately reflects the content of the study, and the abstract is well-structured.

The introduction is well-presented, based on current knowledge about intermittent fasting and weight loss, which allows the objective to be clearly stated.

The methodology is well-structured; however, it mentions that publications are included up to June 2024. It would be advisable to indicate the full time frame, specifying the start date as well as the end. Line 180 mentions that the data collection period spans from 2020 to 2024, which should also be clearly reflected in the abstract within the methodology section.

The results are clearly presented.

The discussion provides a thoughtful reflection on the findings and considers the limitations, such as the fact that only 21 clinical trials could be analysed, among others.

Author Response

Response to Reviewers and Editor:

We are grateful for the helpful comments we received from Reviewers and Editor, which have enabled us to strengthen the scientific merit of our manuscript significantly. Please check the specific revisions in the manuscript, as highlighted in red. Below is the specific response to each comment:

Response to Reviewer 1

This systematic review and meta-analysis is registered with PROSPERO and follows the current guidelines, in accordance with the Preferred Reporting Items for Systematic Reviews and Meta-Analyses (PRISMA) recommendations.

The title accurately reflects the content of the study, and the abstract is well-structured. The introduction is well-presented, based on current knowledge about intermittent fasting and weight loss, which allows the objective to be clearly stated. The methodology is well-structured.

Thank you for your positive feedback on our systematic review and meta-analysis. We are glad to hear that you found the title accurate, the abstract well-structured, and the introduction and methodology clearly presented. We appreciate your insights and look forward to improving the study further.

Comment 1:

however, it mentions that publications are included up to June 2024. It would be advisable to indicate the full time frame, specifying the start date as well as the end.

Thank you for your suggestion. We did not impose a specific start date for the literature search, meaning that publications from the inception of the databases were included. We have included this information in the manuscript. (Ln 77)

Comment 2:

Line 180 mentions that the data collection period spans from 2020 to 2024, which should also be clearly reflected in the abstract within the methodology section.

Thank you for your suggestion. We have added this information to the abstract in the methodology section (Ln 24-25).

Comment 3:

The results are clearly presented.

Thanks. We appreciate your feedback.

Comment 4:

The discussion provides a thoughtful reflection on the findings and considers the limitations, such as the fact that only 21 clinical trials could be analysed, among others.

Thank you so much for all your helpful suggestions, which help us improve the quality of our manuscript.

Reviewer 2 Report

Comments and Suggestions for Authors

It is a systematic review dealing with the effects of time-restricted eating on fat loss in overweight and obese individuals.

This review study analyzed 21 studies which are very difficult to be assessed. There are significant differences in the eating window [from 4 to 12 hours], to the diet prescribed [free eating to calories restricted], to the age of participants [mainly woman], to the combination [or not or not mentioned] of diet with exercise, to the duration of treatment [from 6 to 52 weeks], and so on.

This make  difficult to have a conclusion, although authors suggest that the results provide moderate to high evidence that this form of eating is promizing.

Authors reported a reduction in lean mass; this is attributed to reduced protein intake and to not inclusion of exercise in the all weight-loss protocols. I suggest these to be be mentioned in study limitations

Additionally, the duration of time-restricted intervantion ranges from  one and a half month to 12 months. This should be referred to study limitations - it is imposible to extract results on weight loss and fat loss after 6 weeks and to analyse these findings along with of 12 months treatment.

Minor comments:

I am a little confused with the number of studies reviewed: in detail,

Abstract, line 24 we read: Twenty one publications - 23 studies.

Table 1, 22 studies presented

subsection 3.5.2 TRE protocol: we read in lines 210-212 "The duration of the TRE intervention ranged from 6 to 52 weeks (i.e., 6 (n=3), 7 (n=2), 8 (n=8), 12 (n=4), 14 (n=3), 39 (n=1) 32, and 52 (n=3) weeks)". This is a total of 24 studies.

Lines 212-214 "The eating window length of each TRE was between 4 to 12 hours (i.e., 4 (n=1), 6 (n=2), 8 (n=14), 10 (n=4), and 12 (n=1)hours). In  addition, one study used an 8-hour... " This is a total of 23 studies.

Back to Table 1, eight hours eating window is referred in 12 studies [and not 14 as previously read] and 10 hours window in 6 studies [instead of 4 previously read]. 

What is exactly the truth?

Finally, I would like to see a better editing in Table 1. t omy opinion it is better to be presented in horizontal format. Additionally, I suggest last column [outcomes] to be ommitted and presented in a new Table, where the first column would be the first author's name and then in the next 10 columns the results of the outcomes. The BM, FM, TC, HDL... ect  would be as titles of each column and in each author corresponding line the result, leaving empy the space when no results. By this way they reader will have a clear picture of increase or decrease of each parameter [or not-studied].

Author Response

Response to Reviewers and Editor:

We are grateful for the helpful comments we received from Reviewers and Editor, which have enabled us to strengthen the scientific merit of our manuscript significantly. Please check the specific revisions in the manuscript, as highlighted in red. Below is the specific response to each comment:

Response to Reviewer 2

Comments and Suggestions for Authors

This review study analyzed 21 studies which are very difficult to be assessed. There are significant differences in the eating window [from 4 to 12 hours], to the diet prescribed [free eating to calories restricted], to the age of participants [mainly woman], to the combination [or not or not mentioned] of diet with exercise, to the duration of treatment [from 6 to 52 weeks], and so on.

Comment 1:

This make difficult to have a conclusion, although authors suggest that the results provide moderate to high evidence that this form of eating is promizing.

Thank you for your helpful comments. We have revised our manuscript following them.

Comment 2:

Authors reported a reduction in lean mass; this is attributed to reduced protein intake and to not inclusion of exercise in the all weight-loss protocols. I suggest these to be be mentioned in study limitations

Thanks for pointing out this. We have revised and included more details in this section. (Ln 475-476)

Comment 3:

Additionally, the duration of time-restricted intervantion ranges from one and a half month to 12 months. This should be referred to study limitations - it is imposible to extract results on weight loss and fat loss after 6 weeks and to analyse these findings along with of 12 months treatment.

Thanks for pointing out this. We have revised and included more details in this section. (Ln 463-464)

Comment 4:

Minor comments:

I am a little confused with the number of studies reviewed: in detail,

Abstract, line 24 we read: Twenty-one publications - 23 studies.

Table 1, 22 studies presented

subsection 3.5.2 TRE protocol: we read in lines 210-212 "The duration of the TRE intervention ranged from 6 to 52 weeks (i.e., 6 (n=3), 7 (n=2), 8 (n=8), 12 (n=4), 14 (n=3), 39 (n=1) 32, and 52 (n=3) weeks)". This is a total of 24 studies.

Lines 212-214 "The eating window length of each TRE was between 4 to 12 hours (i.e., 4 (n=1), 6 (n=2), 8 (n=14), 10 (n=4), and 12 (n=1) hours). In addition, one study used an 8-hour... " This is a total of 23 studies.

Back to Table 1, eight hours eating window is referred in 12 studies [and not 14 as previously read] and 10 hours window in 6 studies [instead of 4 previously read]. 

What is exactly the truth?

Thank you for your thorough review, and sorry for the confusion. By carefully checking, we first identified one publication that did not meet the inclusion criteria and thus subsequently excluded it. So for the updated manuscript, 20 publications consisting of 20 studies are included. We have corrected the number accordingly to make it consistent. (Ln144-145)

Comment 5:

Finally, I would like to see a better editing in Table 1. To my opinion it is better to be presented in horizontal format. Additionally, I suggest last column [outcomes] to be ommitted and presented in a new Table, where the first column would be the first author's name and then in the next 10 columns the results of the outcomes. The BM, FM, TC, HDL... etc. would be as titles of each column and in each author corresponding line the result, leaving empty the space when no results. By this way they reader will have a clear picture of increase or decrease of each parameter [or not-studied].

Thank you for your suggestion. We have added a new table (Table 2) as per your recommendation. Further details can be found in that section (Ln 267).